# Self-Reported Gum Bleeding, Perception, Knowledge, and Behavior in Working-Age Hong Kong Chinese—A Cross-Sectional Study

**DOI:** 10.3390/ijerph19095749

**Published:** 2022-05-09

**Authors:** Tsz Yung Wong, Yiu Cheung Tsang, Kim Wai Shadow Yeung, Wai Keung Leung

**Affiliations:** Faculty of Dentistry, The University of Hong Kong, Hong Kong SAR, China; claratywong@connect.hku.hk (T.Y.W.); elvist@hku.hk (Y.C.T.); skwyeung@hku.hk (K.W.S.Y.)

**Keywords:** diagnostic self-evaluation, gingivitis, lifestyle, medicine, Chinese traditional, oral manifestations

## Abstract

Gingivitis and periodontitis are highly prevalent in Hong Kong, where the provision of oral health services is predominantly private. This cross-sectional study aimed to investigate the association between the oral symptoms of gum bleeding and self-reported behavioral factors, beliefs, and knowledge among Hong Kong Chinese. The research team commissioned the Public Opinion Programme of The University of Hong Kong to conduct a structured, population-based, computer-assisted telephone interview (CATI), which covered questions related to the demography, perception, and knowledge (including traditional Cantonese beliefs) of gum health, dental attendance, oral health behavior, and dental anxiety. A total of 1,265 individuals aged 25–60 years old were successfully contacted, and 704 (55.7%) reported prior gum bleeding experience. A total of 516 individuals (64.9% females, median 55–60 years) completed the CATI satisfactorily, and 321 (62.2%) experienced gum bleeding in the past 12 months. The factors that were significantly associated with reports of gum bleeding in the past 12 months include having periodontitis, sensitive teeth, having tertiary or higher education, flossing/interdental cleaning, not cleaning teeth well enough, lack of sleep, consuming too much ‘heaty’ food, avoiding going to the dentist when gums are bleeding, and waiting for gum bleeding to subside (*p* < 0.05, r^2^ =0.198; forward stepwise logistic regression). Within the limitations of this study, approximately half of the Hong Kong working-age adults surveyed reported experiencing gum bleeding, and 62.2% of the participants experienced it within the past 12 months. Members of Hong Kong’s working-age population who reported having higher levels of education appeared more readily aware of their gum problems. Those with bleeding gums, especially those who have discernable periodontitis, poor dental awareness/behaviors, and/or a poor lifestyle should be targeted to receive education and encouragement, which will allow them to take action and improve their own gum health.

## 1. Introduction

Periodontitis is a complex, chronic, inflammatory human disease that affects a significant portion of middle-aged to elderly adults worldwide [1]. Although the nature, etiology, pathogenesis, and treatment approaches of the disease have been well-established, oral healthcare providers are still struggling to reduce the prevalence and severity of this preventable disease globally.

Gingival bleeding is considered a symptom of gingivitis. Persistent gum bleeding, if untreated, can progress to periodontitis, i.e., a disease of the tooth-supporting apparatus that is presentable in various severities and complexities, or ‘Stages,’ with biological features/signs indicative of the progression rate and varying risks, including those related to general health, smoking, etc., demarcated by ‘Grades’ [2]. Bleeding gums relating to periodontal pockets are also easily recognizable to a person affected by periodontitis [3]; however, the symptom is often not sufficient to drive dental attendance [4]. According to the aforementioned survey, only 2.8% of adults with bleeding gums seek professional dental services, while 60% reported neglecting the symptom. In the study, participants reported that ‘treatment not necessary’ and ‘cost’ were common barriers to regular dental attendance.

Because Hong Kong is a southern Chinese city, traditional health beliefs remain an influential part of the oral health concepts of its residents [5,6,7,8,9,10]. An estimated 6–30% of 35–44-year-olds reported that they agreed with the traditional Chinese medicine (TCM) belief that ‘heatiness’ (‘*shang-huo*,’ or ‘*yeet-hay*’ in Cantonese) causes gum disease [5,7,8,10]. Such a belief was found in roughly 30% of the surveyed population in the 1990s [6,7,10]. The concept of heatiness, especially when referring to food, appeared to be prevalent in other Asian cultures and beyond [11,12,13,14,15]. Since heatiness is, in fact, expanding beyond the Asian continent, the regular re-examination of its influence on the oral health of the population is both meaningful and necessary.

Proper knowledge, attitudes, and behaviors are prerequisites for the prevention of gingivitis and periodontal disease. A study in Hong Kong investigated the interactions between oral health knowledge, attitudes, and behaviors, and dental service attendance [16]. The study reported that institutionalized elderly individuals, males, and, interestingly, subjects with better oral health attitudes and behaviors or those without periodontal pockets, were more likely to be irregular dental attendees [16]. This is contrary, however, to an earlier US study that showed that problem-oriented dental attendees had more dental diseases and were less likely to subscribe to dental care [17]. As such, the situation is still somewhat unclear.

This structured, cross-sectional, population-based, computer-assisted telephone interview (CATI) was designed to investigate the factors that might be associated with self-reported gum bleeding in working-age Hong Kong Chinese. The null hypothesis for the current study, concerning self-reported/perceived situations or opinions/actions related to gum bleeding, is as follows: there is no association between gum bleeding in the last 12 months and self-reported gingivitis/periodontitis, oral hygiene practices, awareness/beliefs/practices concerning gum health, dental attendance behavior, pressure, or dental anxiety. The time frame of 12 months for self-reported gum bleeding was adopted to maximize the accuracy/reliability of the self-reported oral conditions [18].

From this section onwards, unless otherwise specified, the phrases ‘gum bleeding’ or ‘bleeding gums’ refer to ‘self-reported gum bleeding.’

## 2. Materials and Methods

### 2.1. Study Design

The project was designed to investigate the self-reported oral health practices and periodontal health beliefs and opinions that are prevalent among working-age Hong Kong Chinese who reported having prior gum bleeding experience via a structured, population-based CATI. The CATI was supported by GlaxoSmithKline (HK) Ltd., Hong Kong SAR, China through Halo Public Relations Ltd., Hong Kong SAR, China. Author WKL, and the Public Opinion Programme (POP) of The University of Hong Kong (HKU) formulated the CATI questions. The study was conducted in Hong Kong from 3–18 April 2014, when the working and actual populations in the city were 3,872,300 and 7,241,700, respectively, distributed across approximately 2.6 million households [19].

### 2.2. Study Participants and Sample Size Calculation

A cross-sectional study design was adopted. Ethical approval was granted by the Human Research Ethics Committee for Non-Clinical Faculties, HKU (ER2014003). To be included, surveyed individuals, according to their own personal statements, had to be 25–60 years old, must have had previous gum bleeding experience, and needed to speak Cantonese.

The survey sample size was, in part, determined in accordance with an earlier CATI study in Hong Kong [20]; i.e., successful cases weighted according to the gender–age distribution of Hong Kong’s Cantonese-speaking (97%) working population of 3,756,131. The regression analysis sample size was determined using Web-Based Sample Size Calculator-Regression [21]. We assumed 8 to 10 final predictors in the concluding multiple logistic regression model and used a common small effect size of 0.20 [22], i.e., 428 to 462 participants would be required in order to yield 85% power at a 5% level of significance. Considering the 10% sample attrition rate and assuming each participant has a 50% probability of having experienced bleeding gums within the past 12 months, at least 1000 telephone contacts were anticipated.

### 2.3. Telephone Survey

The survey was conducted by POP and carried out as described previously [23] under close supervision. In short, random telephone numbers were generated by the CATI system with reference to the first numeric digit used for telephones issued by the Communications Authority, Hong Kong Special Administrative Region. Invalid, facsimile, non-household, call-forwarded, and telephone numbers with technical problems were excluded according to computer and manual dialing records to produce the final sample. Once communication was secured, the member within the household whose next birthday was the closest to the date of the call, and who fulfilled the inclusion criteria, was invited to partake in the survey (Appendix A).

### 2.4. Data Collection

The survey consisted of a five-part questionnaire with 20 questions covering the individual’s history, experiences, and opinions regarding gum bleeding, oral hygiene habits, gum bleeding status, oral healthcare service utilization, dental anxiety (based on a reduced/adapted version [five out of nine questions used] of the Dental Anxiety Inventory short-form [SDAxI]) [24], levels of work-related, financial, and life/family stress, demographic data, and smoking habits (Appendix A).

### 2.5. Data Analysis

The data were analyzed using statistical analysis computer software (IBM SPSS 26.0, IBM, Armonk, NY, USA). To determine the association between the demography of participants, their habits, self-reported gum bleeding management, perceived etiology and/or current dental problems, perceived pressure, dental anxiety, dental attendance, and self-reported gum bleeding over the past 12 months, tests were administered using a Fisher’s exact test, a Chi-square test, and a Student’s *t*-test. Forward stepwise logistic regression analysis was performed to determine whether there was any association between self-reported gum bleeding within the past 12 months [25] and any aforementioned parameter that was identified to be relevant or of interest in general. A *p*-value less than 0.05 was considered statistically significant.

## 3. Results

### 3.1. Participants’ Background

A total of 1,265 individuals aged 25–60 years old were contacted. A total of 704 (55.7%) claimed that they experienced bleeding gums before, and 519 individuals, or 73.7% of the qualified contacts, finished the telephone survey. Three refused or failed to provide an answer for >50% of the survey questions, and hence were excluded in the final analysis. Among the 516 deemed to have completed the survey, 335 (64.9%) were female. The age median was 55–60 years old (Appendix A), or a mean of 44.8 ± 13.4 years (*n* = 507 or 97.7% of participants who reported their exact age, Table 1). Details concerning the demographic background, habits, gum bleeding history/experience, gum bleeding beliefs, dental attendance, dental anxiety, perceived pressure, or periodontal health are summarized in Table 1 and Appendix A. The distributions of age, gender, employment status, and monthly income of the CATI participants are compared to the corresponding year’s census reports (Appendix A). Certain independent variables were recategorized because of the small *n* observed (Appendix A, and Table 1 and Appendix A). The median self-reported work, finances, or life/family pressure scores for the survey participants was 5.0. The reduced/adopted SDAxI questions indicated that 274/217/25 (53.1%/42.1%/4.8%) participants were not at all anxious to somewhat anxious/anxious and tense but were able to control their feelings of anxiety/had extreme anxiety concerning dental treatment, respectively. A similar distribution of dental anxiety levels was observed for participants with and without gum bleeding experience over the past 12 months (data not shown).

Almost all 504 participants (97.7%) reported brushing their teeth with dentifrice every day, while 200 (38.8%) and 135 (26.2%) also reported interdental brushing/flossing or mouth rinsing, respectively. The median claimed that their brushing duration was 2.5 min, while 293 (56.9%) participants self-reported asymptomatic dental attendance [26].

### 3.2. Self-Reported Gum Bleeding Experience, Gum Condition, and Related Health Beliefs

The reported median age at which gum bleeding was first noticed was 16–20 years old. A total of 321 participants (62.2%) reported experiencing gum bleeding over the last 12 months (Table 1). A total of 72 participants (22.4%) from this latter group observed, at least once, that gum bleeding continued for up to three consecutive days (Appendix A). A total of 236 participants (45.8%) reported having sensitive teeth, 173 (33.6%) had receding gums, 158 (30.7%) had painful/swollen gums, and 91 (17.7%) reported having periodontitis over the CATI (Table 1).

Concerning situations considered to be associated with bleeding gums, 194 participants (39.0%) reported having poor oral hygiene, while 154 (29.8%), 135 (26.2%), 91 (17.6%), and 65 (12.6%) reported excessive heaty food intake, lacking sleep/late-night activities, poor general health/sickness, and being in a bad mood or stressed, respectively (Table 1).

Regarding the perceived cause of their bleeding gums, 471 participants (91.3%) reported that it was due to poor oral hygiene, gingivitis, or periodontitis; however, 241 (46.7%), 221 (42.8%), 105 (20.3%), 100 (19.3%), and 64 (19.1%, females only) reported that it could also be caused by heaty food consumption, lacking sleep/late-night activities, being in in bad mood or stressed, smoking, and pregnancy, respectively (Table 1).

A total of 137 participants (26.6%) and 255 participants (49.4%) considered gingivitis and periodontitis, respectively, to be the sequelae if gum bleeding was ignored. However, at least 18 participants (3.5%) reported that they had no idea about the outcomes of ignoring gum bleeding (Appendix A).

### 3.3. Self-Reported Actions Taken for Gum Bleeding

Regarding the remedy for bleeding gums, an alarming 387 participants (75.0%) said that they ignored the condition and hoped for ‘spontaneous recovery’. A total of 211 (40.9%) reported that they tried to improve their oral hygiene, while only 114 (22.1%) said that they would seek dental care. A total of 275 participants (53.3%) used a saline mouth rinse, 228 (44.2%) reported taking Chinese herbal medicine/cutting down on heaty food, and 139 (26.9%) avoided brushing the region that bled (Table 1).

### 3.4. Factors Associated with Self-Reported Gum Bleeding over the Past 12 Months

Forward stepwise logistic regression was performed in relation to self-reported gum bleeding over the past 12 months, and relevant factors were screened in the CATI (Table 1 and Table 2). From the results, no significant association could be observed between the self-reported gum bleeding experiences over the past 12 months and smoking habits, employment status, monthly income, dental attendance, dental anxiety, age when gum bleeding was noticed, or perceived outcomes of gum bleeding not being properly attended to (Appendix A). Regarding the perceived pressure level, the variable ‘self-perceived pressure at work’ was considered eligible (*p* = 0.087, Table 1) to be included in the logistic regression model; however, bearing in mind that only 361 out of 516 participants (70.0%) worked part-/full-time, to prevent a significant drop in sample size, the variable was not included in the final logistic regression analysis.

The logistic regression analysis showed that the survey respondents who claimed to currently have periodontitis, had sensitive teeth, had tertiary or higher education, practiced flossing/interdental cleaning, had related gum bleeding experience due to not cleaning teeth well, suffered from lack of sleep, consumed too much heaty food, ignored the problem, and awaited ‘spontaneous healing’ rather than visiting the dentist were significantly associated with reporting bleeding gums in the past 12 months (*p* < 0.05, r^2^ =0.198; Table 2).

## 4. Discussion

This study evaluated the self-reported gum condition of working-age Hong Kong southern Chinese using CATI—a well-proven, non-face-to-face survey methodology [27]. It is worth noting that only a few large-scale studies/trials [28,29] have investigated self-reported gum bleeding; thus, the findings from this current study can perhaps fill this knowledge gap, especially concerning the perception and knowledge of gum health, oral health behaviors, and dental anxiety. This study attempted to characterize, in a patient-centered fashion, how the condition and experiences of bleeding gums were appreciated with respect to demography, oral hygiene practices, periodontal health awareness, related health beliefs and behaviors, perceived pressure levels, dental attendance, and home management strategies. In particular, this study explored concepts including Asian/Chinese beliefs regarding gum bleeding; e.g., heaty food consumption. The above information forms part of the indispensable building blocks for successful, prevention-oriented oral health education/care [30].

In the present study, almost one-fifth of the participants reported having periodontitis, which was in line with the clinical observations derived from a local cross-sectional study by the same group [31]. Therefore, the present self-reported periodontitis, as reported earlier [32], can be considered accurate.

In addition, 61.8% of those surveyed reported experiencing gum bleeding over the past 12 months, while 56.8% had asymptomatic dental attendance (Appendix A). These results are similar to those of the Oral Health Survey based on the Hong Kong local population in 2001 [8] and 2011 [10].

Regarding the reported actions taken to manage bleeding gums, from the 2011 Oral Health Survey, 61.2% claimed no action was taken and 31.7% ‘self-managed’ the condition [10]. The current cohort reported that 75% of them preferred to ignore the condition and wait for ‘spontaneous recovery.’ A higher percentage of this study’s survey participants reported seeking dental care for bleeding gums (22.2%) and self-management by improving oral hygiene (41%), compared to 7.0% and 31.7% in the 2011 Oral Health Survey, respectively [10]. This present study, however, allowed participants to provide multiple responses to the questions, which perhaps accounts for the discrepancies between studies.

Previous studies reported that factors such as daily stress might affect dental behaviors, including avoidance of regular dental attendance, and positively relate to self-reported gum bleeding [29,33]. In contrast to these studies, perceived financial and/or life/family pressure was not associated with self-reported gum bleeding over the past 12 months (Table 1) in the current study. Nevertheless, we did not attempt to correlate perceived work pressure (*p* = 0.087, Table 1) and self-reported gum bleeding in the forward regression analysis because 30.0% of participants were not working (Table 1). Further investigations focusing exclusively on working adults are needed to clarify if there is any association between self-perceived pressure from work and self-reported gum bleeding.

The current cohort appeared to exhibit relatively similar dental anxiety levels compared to those reported earlier [34]; i.e., approximately 10% or fewer of survey participants had a high level of dental anxiety. A report by Marques-Vidal and Milagre [35] indicated that hospital anxiety was not associated with self-reported bleeding gums, which is in line with this current study (Appendix A). On the contrary, a study concerning rural older American adults [36] reported an association between responses to individual dental anxiety questions and self-reported gum bleeding.

The concept of heatiness, and particularly heaty or cold foods, is deeply embedded in Hong Kong and other Asian cultures [12,13]. In short, the heatiness of food depends on the number of calories and the amount of fat (i.e., the higher the number, the heatier the food), as well as the ingredients used (e.g., chili, ginger, etc.) and the way in which the food is prepared (e.g., deep-fried). Otherwise, non-heaty dishes can be turned into heaty ones, and the heatiness of already heaty ingredients can be intensified [13]. In a previous study, 623 of 957 subjects (65%) reported a moderate to strong Chinese oral health belief that heatiness causes gum disease at a general and/or personal level, while 280 reported that the consumption of ‘*xie-huo*’ (‘fire purging’) herbal tea or other forms of TCM helped to prevent or self-manage swollen or bleeding gums [7]. Overall, 361 (38%) claimed to use Chinese preventive practices to maintain their oral health.

Bleeding gums, or ‘*ya-nu*’ in the TCM context [14], indicates poor oral hygiene associated with an imbalance in the human body in terms of ‘*stomach-heat*,’ ‘*kidney-yin deficiency*,’ and ‘*weakness in qi and blood*’ [37].

According to the TCM concept of heatiness, the affected person often displays signs of stress fatigue syndrome, i.e., a combination of dryness and a bitter taste in the mouth, mouth ulcers, halitosis, gingival swelling, sore throat, low fever, insomnia, yellow urine, constipation, backache, lack of appetite, fatigue, nervousness, and anxiety [34]. Gum bleeding is thus an easily recognizable symptom of heatiness. Cantonese people who are affected by stress-fatigue syndrome and who also embrace the TCM concept of heatiness look to reduce their workload, increase rest, reduce stress, and consume *xie-huo* TCM or food to rectify the situation [15,38].

Our cohort, as previously reported [7], was somewhat confused about gum bleeding, since a large proportion of those who reported gum bleeding over the past 12 months believed that it was associated with the consumption of heaty food. As observed in an earlier study [11], although the present cohort appeared to have strong traditional/conventional Chinese oral health beliefs, none recognized the fact that the TCM concept of gum bleeding also embraces good oral home care, in addition to maintaining ‘*yin-yang*’—a balance of body and lifestyle [14,39,40]. The association of poor self-reported general health with gingival bleeding was also reported in a Finnish study [41].

Despite those surveyed realizing their bleeding gums were associated with poor/inadequate oral hygiene (39%), or poor health behavior being considered the cause of the bleeding gums (91%, Table 1), only 41% reported that they had attempted to improve their oral hygiene. Considering the inconsistencies between proper health beliefs and a lack of appropriate self-care actions [42], we suspect that the participants may lack the knowledge of how to manage oral hygiene [43]. At the same time, 75% reported not doing anything about gingival bleeding. We suspect such negligence stems from poor self-efficacy [44]; however, the reason cannot be determined in this study due to its cross-sectional nature.

Logistic regression analysis found that self-reported ‘not brushing frequently and/or cleaning teeth well enough’ was a significant parameter that was associated with a history of gum bleeding over the past 12 months (Table 2). Paradoxically, however, self-reported use of floss/interdental brushing was associated with gum bleeding. Due to the self-reported, cross-sectional nature of this report, it is not possible for us to explore the reasons for these inconsistencies. That said, a Finnish prospective study reported that people who were aware of their own bleeding gums appeared to be a positive predictive factor for subsequent gingival health improvement among the cohorts that followed [45]. We suspect a similar influence on the self-reported oral health behavior of the participants who completed the CATI.

It is intriguing to note that higher education was significantly associated with self-reported gum bleeding over the past 12 months (Table 2). A similar finding was reported in a large-scale global study [27]. However, that particular study did not elaborate on or interpret the association. We suspect that those participants with higher education are more health-conscious and thus better able to recognize gum bleeding in the past 12 months.

Although there has yet to be a study regarding the effects of a lack of sleep or sleeplessness on gum bleeding in working adults, a recent Korean national survey targeting the effects of problematic Internet usage among adolescents (*n* = 73,238, 13–18 years old) reported that the high-risk group, i.e., 5–6 h of Internet use per day, was associated with higher odds for the oral symptoms of sore and bleeding gums [46]. In the reported path analysis, problematic Internet usage appeared to affect sleep and indirectly affected oral health [46]. Further study, however, is needed to determine if a lack of sleep might also be associated with gum bleeding in working adults.

Regarding the excess consumption of heaty food and gum bleeding, a cross-sectional study investigated 96 Chinese submarine military personnel. Under a calorie-sufficient meat, vegetable, and oil-based diet, their BMI, body fat, and self-reported symptoms were suggestive of improper nutrition [47]. A significant number of subjects (23%) reported gum bleeding. The high-fat, heaty food, as well as the confined, stressful lifestyle, could perhaps have induced stress fatigue syndrome. We assume a similar situation among the participants of this present study who experienced bleeding gums after consuming heaty food, i.e., a risk behavior associated with stress fatigue syndrome.

An equally interesting observation was that participants who reported not visiting a dentist for gum bleeding were associated with having bleeding gums over the past 12 months (Table 2). A telephone survey by Young [48] found that misunderstandings in Hong Kong regarding dental scaling appeared very prevalent, regardless of whether the individual had experienced scaling or not. This could explain why only around 20% of participants in this present study reported seeking dental care after experiencing gum bleeding (Table 1 and Table 2). It is worth noting that private dental practitioners are the primary oral health care providers in Hong Kong [49], and local dentists exhibited the highest pro-rich inequality for utilization among the three high-income Asian economies of South Korea, Taiwan, and Hong Kong [50]. A study on the dental attendance of middle-aged to elderly people in Hong Kong revealed that the majority of dentist visits were pain-driven, with the primary reason for non-attendance being that they were too busy [51]. Usually, bleeding gums are not painful [52], which perhaps explains why a relatively low proportion of the CATI participants reported going to the dentist. Further investigation and intervention are needed to address this disappointing oral care utilization behavior in Hong Kong.

The results of this study rejected the null hypothesis, i.e., self-reported inadequate oral hygiene practice, poor awareness concerning gum health, and poor dental attendance behavior were associated with self-reported gum bleeding over the past 12 months. In addition, under the influence of Chinese oral health beliefs, consuming too much heaty food and having a lack of sleep are also perceived to be associated factors. Under the strong influence of the TCM *shang-hou* concept, despite not being at a statistically significant level, participants also reported the consumption of herbal medicine in the management of bleeding gums, in addition to brushing teeth, flossing, and/or mouth rinsing. The above factors need to be taken into consideration when planning for oral health promotion and education, preventing bleeding gums, and caring for the local/Asian population, particularly with the aim of rectifying incorrect oral health beliefs and improper oral health behaviors.

Caution should be exercised when interpreting the data of this present cross-sectional study due to a number of issues/limitations. First, only participants with a self-reported history of gum bleeding were recruited; those who never experienced bleeding gums and their oral health behavior/awareness/beliefs/practices/dental attendance, etc., were not included nor explored in the current data analysis. Second, social desirability and recall bias are common problems associated with studies that use questionnaires for data collection [53]. Third, we did not investigate the antibiotic usage history of the participants. Thus, we were unable to exclude the potential influence of medication on the possible attenuation of gum bleeding. Fourth, except for prior information given by oral healthcare providers concerning periodontitis/deep pockets status [54], self-reported periodontal condition, despite being considered relatively valuable, is less reliable [18]. Fifth, survey participants might report gum bleeding due to other factors such as trauma (e.g., a broken tooth, poor margins, or traumatic oral hygiene) or health problems (e.g., poor nutrition, systemic disease, or medication). That being said, gingivitis/periodontitis was still the main reason for gum bleeding [55]. Sixth, to ensure the high clinical accuracy of the survey data, a follow-up examination of the participants is needed. However, this is not commonly a part of the CATI design and was thus not implemented for this study. Seventh, due to the nature of cross-sectional studies/surveys, the current results cannot be considered generalizable without proper design and the execution of replication studies in the same population and beyond.

## 5. Conclusions

Despite the limitations of this study as described above, a number of factors were identified to be associated with self-reported gum bleeding. These include currently having periodontitis, sensitive teeth, having tertiary or higher education, using floss or an interdental brush, having related gum bleeding experience due to not cleaning teeth well, a lack of sleep, consuming too much heaty food, preferring not to consult a dentist about the problem, and preferring to wait for spontaneous resolution of gum bleeding. To promote periodontal health among the public, the determinants underpinning insufficient individual knowledge or improper oral health behaviors must be identified and rectified by all parties concerned. Dentists should be proactive and seriously undertake patient-centered therapy in an attempt to rectify misconceptions and promote proper oral health behaviors and dental attendance among the Hong Kong population.

## Figures and Tables

**Table 1 ijerph-19-05749-t001:** Background, habit, gum bleeding experience, and perceptions of study participants (*n* = 516).

	Gum Bleeding over Last 12 Months?				*p*-Value ^1^
	No (*n* =195)		Yes (*n* = 321)		
**Age (year; mean, SD)** (No, *n* = 191, Yes, *n* = 316)	45.24	±10.52	44.88	±10.39	0.706
**Gender**					
- Male	68	(34.9%)	113	(35.2%)	0.939
- Female	127	(65.1%)	208	(64.8%)	
**Education**					
- Secondary or below	144	(73.8%)	203	(63.2%)	0.017
- Tertiary or above	51	(26.2%)	118	(36.8%)	
**Perceived pressure level (0–10 Likert scale; 0 = No, 10 = Max.; mean)**					
- Work-related (limited to respondents who are working; No, *n* = 136, Yes, *n* = 225)	4.43		4.85		0.087 ^2^
- Financial	3.77		4.07		0.191
- Life/family	3.59		3.81		0.322
**Oral hygiene practice** ^3^					
- Brushing with toothpaste	188	(96.4%)	316	(98.4%)	0.237
- Using floss/interdental brush	66	(33.8%)	134	(41.7%)	0.091
- Using a mouth rinse	55	(28.2%)	80	(24.9%)	0.472
- Brushing with toothpaste + floss/interdental brush	65	(33.3%)	133	(41.4%)	0.082
- Brushing with toothpaste + floss/interdental brush + mouth rinse	16	(8.2%)	37	(11.5%)	0.291
- Brushing ≥ 2 min. after waking up in the morning	127	(65.1%)	211	(65.7%)	0.965
- Brushing ≥ 2 min. before going to bed	122	(62.6%)	203	(63.2%)	0.952
- Brushing ≥ 2 min. after waking up and in the evening before going to bed	116	(59.5%)	190	(59.2%)	0.947
**Experienced gum bleeding under situations when:** ^3^					
- Too much heaty food was consumed (e.g., hot pot)	34	(17.4%)	120	(37.4%)	<0.001
- In poor general health/fell sick (e.g., flu, common cold, sore throat, on medication)	20	(10.3%)	71	(22.1%)	<0.001
- Feeling irritated/under pressure	13	(6.7%)	52	(16.2%)	0.003
- Lack of sleep/going through the night(s) without sleeping	27	(13.8%)	108	(33.6%)	<0.001
- Not brushing frequently and/or cleaning teeth well enough (No, *n* = 188, Yes, *n* = 309)	47	(25.0%)	147	(47.6%)	<0.001
**Perceived cause(s) of own gum bleeding** ^3^					
- Consuming heaty food	83	(42.6%)	158	(49.2%)	0.168
- Feeling irritated/under pressure	38	(19.5%)	67	(20.9%)	0.736
- Lack of sleep/going through the night(s) without sleeping	70	(35.9%)	151	(47.0%)	0.014
- Smoking	35	(17.9%)	65	(20.2%)	0.567
- Poor oral hygiene/having gingivitis/having periodontitis	171	(87.7%)	300	(93.5%)	0.235
- Being pregnant (limited to females; No, *n* = 127, Yes, *n* = 208)	29	(22.8%)	35	(16.8%)	0.225
**Managed own bleeding gums by:** ^3^					
- Attending dentist	56	(28.7%)	58	(18.1%)	0.005
- Brushing better/using floss/interdental brush	71	(36.4%)	140	(43.6%)	0.128
- Ignoring, awaiting ‘spontaneous healing’	128	(65.6%)	259	(80.7%)	<0.001
- Consuming herbal tea to purge the heat/stop eating heaty food	79	(40.5%)	149	(46.4%)	0.223
- Rinsing mouth with saline	106	(54.4%)	169	(52.6%)	0.774
- Avoiding the bleeding sites while brushing	60	(30.8%)	79	(24.6%)	0.154
**Current dental problem(s):**^3^ (No, *n* = 195, Yes, *n* = 320)					
- Periodontitis	16	(8.2%)	75	(23.4%)	<0.001
- Mobile/drifting teeth	27	(13.8%)	74	(23.1%)	0.014
- Receding gums/teeth became longer	50	(25.6%)	123	(38.4%)	0.004
- Swollen/painful gums	44	(22.6%)	114	(35.6%)	0.003
- Sensitive teeth	64	(32.8%)	172	(53.8%)	<0.001
- Had extraction/exfoliation of teeth due to periodontitis	15	(7.7%)	34	(10.6%)	0.345

^1^ Fisher’s Exact Test/Chi-square Test/*t*-test. ^2^ Not included in the forward stepwise logistic regression analysis (Table 2) because 155 (30.0%) of participants were not working at the time of the survey (Appendix A). ^3^ Multiple answers were allowed, so the total does not add up.

**Table 2 ijerph-19-05749-t002:** Logistic Regression analysis of variables associated with self-reported gum bleeding over past 12 months (No = 0, Yes = 1).

	Unadjusted				Adjusted			
Variables ^1^	Coefficient	*p*-Value	95% CI		Coefficient	*p*-Value	95% CI	
Constant	−1.602	0.027			−1.087	<0.001		
Age	0.002	0.836	0.980	1.025				
Gender	0.162	0.476	0.754	1.833				
Education (secondary or below/tertiary or above)	0.520	0.038	1.030	2.746	0.450	0.048	1.005	2.451
*Oral hygiene practice*								
Using floss/interdental brush	0.287	0.859	0.055	32.076	0.469	0.030	1.048	2.440
Brushing with toothpaste + floss/interdental brush	0.166	0.919	0.049	28.607				
*Experienced gum bleeding when*								
Too much *heaty* food was consumed	0.731	0.004	1.257	3.434	0.772	0.002	1.318	3.550
in poor general health/fell sick	0.503	0.112	0.890	3.070				
Feeling irritated/under pressure	−0.045	0.909	0.443	2.064				
Lack of sleep/going through the night(s) without sleeping	0.633	0.034	1.048	3.384	0.719	0.009	1.197	3.517
Not brushing frequently and/or cleaning teeth well enough	0.816	<0.001	1.446	3.535	0.779	<0.001	1.409	3.372
*Managed own bleeding gums by*								
Attending dentist	−0.997	<0.001	0.224	0.608	−1.007	<0.001	0.223	0.597
Ignoring, awaiting ‘spontaneous healing’	0.613	0.012	1.144	2.977	0.580	0.014	1.125	2.835
*Current dental problem*								
Periodontitis	0.948	0.006	1.304	5.105	1.081	0.001	1.567	5.550
Mobile/drifting teeth	0.208	0.501	0.672	2.255				
Recedings gum/teeth became longer	0.032	0.898	0.635	1.679				
Swollen/painful gums	0.197	0.425	0.750	1.978				
Sensitive teeth	0.543	0.015	1.113	2.662	0.634	0.003	1.245	2.853

r^2^ (0.198) was computed using forward stepwise logistic regression with variables that were significant (*p* < 0.05). ^1^ ‘Perceived cause(s) of own gum bleeding: lack of sleep/going through the night(s) without sleeping’ was excluded in the final analysis because of the likelihood of confounding it with ‘Experienced gum bleeding when: lack of sleep/going through the night(s) without sleeping’; reference for gender: male.

## Data Availability

The data (Chinese version) that supports the findings of this study are available on: https://www.hkupop.hku.hk/english/report/gumBleeding2014/index.html (accessed on 10 March 2022).

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
