# Peer review of "Self-Reported Gum Bleeding, Perception, Knowledge, and Behavior in Working-Age Hong Kong Chinese—A Cross-Sectional Study"

_ijerph, 2022, doi:10.3390/ijerph19095749_

Round 1

Reviewer 1 Report

The authors carried out a survey on a cohort of the Hong Kong population, using the surveyed prior experience on gum bleeding as a means of studying their perception, knowledge and behaviors related to gingivitis and periodontitis. The reviewer finds the study comprehensive.

Please see additional comments below.

The authors carried out a survey on a cohort of the working age Hong Kong Chinese population, using the surveyed prior experience on gum bleeding as a means of studying their perception, knowledge and behaviors related to gingivitis and periodontitis. Previous studies on the topic were reported prior to 2016 and the present study exclusively focus on gingivitis and periodontitis based on a questionnaire relevant to the primary symptom of the disease, gum bleeding.

The study provides the most recent information on the research area that has been conducted on a such a cohort and provide in depth information. Questions covering topics including demography, perception and knowledge of gum health, dental attendance, oral health behavior and dental anxiety makes it a comprehensive study covering the Hong Kong Chinese population.

Although there are previous studies on gum bleeding reported, the current study would answer many underlying questions such as bleeding gum experience, demography, oral hygiene practices, behavior, perceived pressure level, periodontal health awareness, bleeding gum related beliefs and management strategies which provides important information related to periodontitis and gingivitis.

The study explore concepts including Asian/Chinese beliefs on gum bleeding particularly 'heaty food' consumption. Attempt are made to understand relationship between oral hygiene, stress, smoking and pregnancy. Overall, the study provides a comprehensive and more recent dataset/information on the topic.

 Should any participants treated with antibiotic (particularly for other health conditions within the past 12 months) be excluded from the study? Antibiotic treatment may have impacted gum bleeding and the survey questions 4,5 and 6 might be affected.

The conclusions are consistent with the results. The references are appropriate

Line 57: It would be better to cite a paper with contradicting observations (to ref #10), to make a statement that the "field remains rather confused". Line 366: Please mention what are the limitations in the study.

Reviewer 2 Report

This cross-sectional study assessed the self-perception, knowledge and behaviors of 25-60-year old individuals living in Hong Kong in relation to their gingival bleeding. Data were recorded by means of a structured telephone interview. Although this topic ha been previously investigated, this study aims to provide some new insights on the Chinese population. English language should be carefully revised.

Major comments:

The abstract and the paper need to be better reorganized and to be extensively revised. Many sentences are difficult to read and the discussion section should be shortened (the first paragraphs can be removed).

The study lacks of a clearly stated research hypothesis. The methods of participants’ selection should be better described. Was this sample representative of the whole adult Hong Kong working population? The sample size calculation should be described in details.

The authors should report more information in the Material and Methods section about the parameters they collected.

The selection criteria were not clearly described. The authors stated that participants had to be 25-60-year old, speaking Cantonese and having gingival bleeding experience. However, in the Table 1 and in the Results section they were stratified on having or not gum bleeding in the last 12 months.

The statistical analysis should be improved. Please, reconsider the variables put in the univariate and multiple logistic regression models. I wonder whether the variables related the daily situations in which participants experienced gum bleeding (feeling under pressure, feeling sick, lacking sleep) and those related to the perceived cause of gum bleeding (lack of sleep, feeling under pressure…..) can be predictors of the dependent variable (self-perception of gum bleeding over the last 12 months). In addition, most of them overlap.

Minor comments:

Many sentences should be revised (i.e. lines 12-15, lines 18-23, lines 34-35, lines 42, lines 43-46….). The sentence in lines 64-65 should be moved in the Material and Methods section. Please remove the column “yes-no” from Table 1. Please add the number of the Ethical Committee approval.

Reviewer 3 Report

Need to make small modifications. It would be convenient a specific description of periodontitis, updated with the latest worshop published.

1. What is the main question addressed by the research?

It is a telephone survey at the Hong Kong Chinese  population, on whether they know if their gums bleed or not, like a self-diagnosis to validate the incidence of this problem in that global population.

2. Do you consider the topic original or relevant in the field, and if
so, why?

it is not a relevant topic, much less original. It is a topic very treated before and little relevant also the population can be conditioned, they aren't profesional. Furthermore different aspects can be the gum bleeding reason, and that factor is not mention in the paper.

3. What does it add to the subject area compared with other published
material?

It does not aport anything relevant aspect. Is a very treated topic for many years and it only show a very simple par, the self-diagnosis without treatment or follow-up.

4. What specific improvements could the authors consider regarding the
methodology?

Some aspects can be improve, for example: The bleeding gums diagnosis by phone can be confirmed by a periodontal professional who can also says the reason of that bleeding, due to inflammation and not trauma or damage.

5. Are the conclusions consistent with the evidence and arguments
presented and do they address the main question posed?

Yes, they are adequate.

6. Are the references appropriate?

Yes, are correct.

7. Please include any additional comments on the tables and figures.

No changes are necessary.

Round 2

Reviewer 2 Report

The research hypothesis needs to be more focused and the terms should be used more consistently (gum bleeding or perceived gum bleeding?). What is the primary objective of the study? Did the authors perform a priori the sample size calculation? What are the underlying assumptions (effect size, parameter)? How did the sampling method assure the representativeness of the working age Hong Kong population?  

In a cross-sectional study design a representative sample of the target population is consecutively selected and then stratified according to the outcome and the exposure. In this study the authors enrolled only subjects who reported suffering from gum bleeding and then they divided them into two groups according to the time in which they experienced gum bleeding (more or less than 12 months). The authors quoted a study published in 2007 to support their sampling strategy. However, this survey enrolled a sample of 507 subjects to assess the community’s level of tolerance towards mental illness.  
